# Parallel Recursive Best-First AND/OR Search for Exact MAP Inference in Graphical Models

**Akihiro Kishimoto**
IBM Research, Ireland
akihirok@ie.ibm.com

**Radu Marinescu**
IBM Research, Ireland
radu.marinescu@ie.ibm.com

**Adi Botea**
IBM Research, Ireland
adibotea@ie.ibm.com

## Abstract

The paper presents and evaluates the power of *parallel search* for exact MAP inference in graphical models. We introduce a new parallel shared-memory recursive best-first AND/OR search algorithm, called SPRBFAOO, that explores the search space in a best-first manner while operating with restricted memory. Our experiments show that SPRBFAOO is often superior to the current state-of-the-art sequential AND/OR search approaches, leading to considerable speed-ups (up to 7-fold with 12 threads), especially on hard problem instances.

## 1 Introduction

Graphical models provide a powerful framework for reasoning with probabilistic information. These models use graphs to capture conditional independencies between variables, allowing a concise knowledge representation and efficient graph-based query processing algorithms. Combinatorial maximization, or maximum *a posteriori* (MAP) tasks arise in many applications and often can be efficiently solved by search schemes, especially in the context of AND/OR search spaces that are sensitive to the underlying problem structure [1].

*Recursive best-first AND/OR search* (RBFAOO) is a recent yet very powerful scheme for exact MAP inference that was shown to outperform current state-of-the-art depth-first and best-first methods by several orders of magnitude on a variety of benchmarks [2]. RBFAOO explores the context minimal AND/OR search graph associated with a graphical model in a *best-first* manner (even with non-monotonic heuristics) while running within restricted memory. RBFAOO extends Recursive Best-First Search (RBFS) [3] to graphical models and thus uses a threshold controlling technique to drive the search in a depth-first like manner while using the available memory for caching.

Up to now, search-based MAP solvers were developed primarily as sequential search algorithms. However, parallel, multi-core processing can be a powerful approach to boosting the performance of a problem solver. Now that multi-core computing systems are ubiquitous, one way to extract substantial speed-ups from the hardware is to resort to parallel processing. Parallel search has been successfully employed in a variety of AI areas, including planning [4], satisfiability [5], and game playing [6, 7]. However, little research has been devoted to solving graphical models in parallel. The only parallel search scheme for MAP inference in graphical models that we are aware of is the *distributed AND/OR Branch and Bound* algorithm (daoopt) [8]. This assumes however a large and distributed computational grid environment with hundreds of independent and loosely connected computing systems, without access to a shared memory space for caching and reusing partial results.

**Contribution** In this paper, we take a radically different approach and explore the potential of parallel search for MAP tasks in a *shared-memory environment* which, to our knowledge, has not been attempted before. We introduce SPRBFAOO, a new parallelization of RBFAOO in shared-memory environments. SPRBFAOO maintains a single cache table shared among the threads. In this way, each thread can effectively reuse the search effort performed by others. Since all threads start from the root of the search graph using the same search strategy, an effective load balancing is

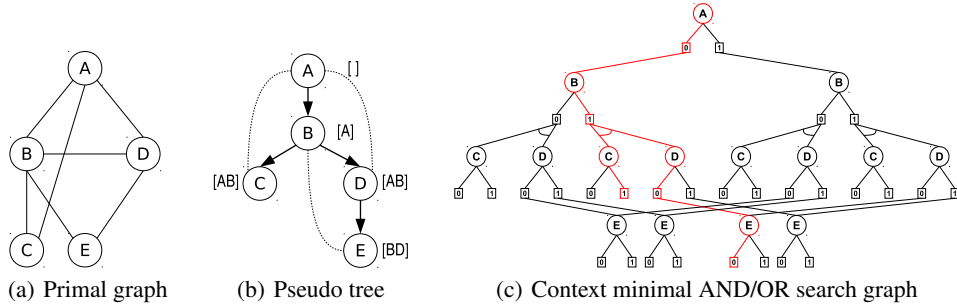

| (a) Primal graph | (b) Pseudo tree | (c) Context minimal AND/OR search graph |

Figure 1: A simple graphical model and its associated AND/OR search graph.

obtained without using sophisticated schemes, as done in previous work [8]. An extensive empirical evaluation shows that our new parallel recursive best-first AND/OR search scheme improves considerably over current state-of-the-art sequential AND/OR search approaches, in many cases leading to considerable speed-ups (up to 7-fold using 12 threads) especially on hard problem instances.

## 2 Background

*Graphical models* (e.g., Bayesian Networks [9] or Markov Random Fields [10]) capture the factorization structure of a distribution over a set of variables. A *graphical model* is a tuple $\mathcal{M} = \langle \mathbf{X}, \mathbf{D}, \mathbf{F} \rangle$, where $\mathbf{X} = \{X_i \ : \ i \in V\}$ is a set of variables indexed by set $V$ and $\mathbf{D} = \{D_i \ : \ i \in V\}$ is the set of their finite domains of values. $\mathbf{F} = \{\psi_\alpha \ : \ \alpha \in F\}$ is a set of discrete positive real-valued local functions defined on subsets of variables, where $F \subseteq 2^V$ is a set of variable subsets. We use $\alpha \subseteq V$ and $\mathbf{X}_\alpha \subseteq \mathbf{X}$ to indicate the *scope* of function $\psi_\alpha$, i.e., $\mathbf{X}_\alpha = var(\psi_\alpha) = \{X_i \ : \ i \in \alpha\}$. The function scopes yield a *primal graph* whose vertices are the variables and whose edges connect any two variables that appear in the scope of the same function. The graphical model $\mathcal{M}$ defines a factorized probability distribution on $\mathbf{X}$, as follows: $P(\mathbf{X}) = \frac{1}{Z} \prod_{\alpha \in F} \psi_\alpha(\mathbf{X}_\alpha)$ where the *partition function*, $Z$, normalizes the probability.

An important inference task which appears in many real world applications is *maximum a posteriori* (MAP, sometimes called *maximum probable explanation* or *MPE*). MAP/MPE finds a complete assignment to the variables that has the highest probability (i.e., a mode of the joint probability), namely: $\mathbf{x}^* = \mathrm{argmax}_{\mathbf{x}} \prod_{\alpha \in F} \psi_\alpha(\mathbf{x}_\alpha)$ The task is NP-hard to solve in general [9]. In this paper we focus on solving MAP as a *minimization* problem by taking the negative logarithm of the local functions to avoid numerical issues, namely: $\mathbf{x}^* = \mathrm{argmin}_{\mathbf{x}} \sum_{\alpha \in F} -\log(\psi_\alpha(\mathbf{x}_\alpha))$.

Significant improvements for MAP inference have been achieved by using AND/OR search spaces, which often capture problem structure far better than standard OR search methods [11]. A *pseudo tree* of the primal graph captures the problem decomposition and is used to define the search space. A pseudo tree of an undirected graph $G = (V, E)$ is a directed rooted tree $\mathcal{T} = (V, E')$, such that every arc of $G$ not included in $E'$ is a back-arc in $\mathcal{T}$, namely it connects a node in $\mathcal{T}$ to an ancestor in $\mathcal{T}$. The arcs in $E'$ may not all be included in $E$.

Given a graphical model $\mathcal{M} = \langle \mathbf{X}, \mathbf{D}, \mathbf{F} \rangle$ with a primal graph $G$ and a pseudo tree $\mathcal{T}$ of $G$, the *AND/OR search tree* $S_\mathcal{T}$ has alternating levels of OR nodes corresponding to the variables and AND nodes corresponding to the values of the OR parent's variable, with edges weighted according to $\mathbf{F}$. We denote the weight on the edge from OR node $n$ to AND node $m$ by $w(n, m)$. Identical sub-problems, identified by their *context* (the partial instantiation that separates the sub-problem from the rest of the problem graph), can be merged, yielding an *AND/OR search graph* [11]. Merging all context-mergeable nodes yields the *context minimal AND/OR search graph*, denoted by $C_\mathcal{T}$. The size of $C_\mathcal{T}$ is exponential in the induced width of $G$ along a depth-first traversal of $\mathcal{T}$ [11].

A *solution tree* $T$ of $C_\mathcal{T}$ is a subtree such that: (1) it contains the root node of $C_\mathcal{T}$; (2) if an internal AND node $n$ is in $T$ then all its children are in $T$; (3) if an internal OR node $n$ is in $T$ then exactly one of its children is in $T$; (4) every tip node in $T$ (i.e., nodes with no children) is a terminal node. The cost of a solution tree is the sum of the weights associated with its edges.

Each node $n$ in $C_{\mathcal{T}}$ is associated with a *value* $v(n)$ capturing the optimal solution cost of the conditioned sub-problem rooted at $n$. It was shown that $v(n)$ can be computed recursively based on the values of $n$'s children: OR nodes by minimization, AND nodes by summation (see also [11]).

**Example 1.** *Figure 1(a) shows the primal graph of a simple graphical model with 5 variables and 7 binary functions. Figure 1(c) displays the context minimal AND/OR search graph based on the pseudo tree from Figure 1(b) (the contexts are shown next to the pseudo tree nodes). A solution tree corresponding to the assignment $(A = 0, B = 1, C = 1, D = 0, E = 0)$ is shown in red.*

Current state-of-the-art sequential search methods for exact MAP inference perform either depth-first or best-first search. Prominent methods studied and evaluated extensively are the *AND/OR Branch and Bound* (AOBB) [1] and *Best-First AND/OR Search* (AOBF) [12]. More recently, *Recursive Best-First AND/OR Search* (RBFAOO) [2] has emerged as the best performing algorithm for exact MAP inference. RBFAOO belongs to the class of RBFS algorithms and employs a local threshold controlling mechanism to explore the AND/OR search graph in a depth-first like manner [3, 13]. RBFAOO maintains at each node $n$ a lower-bound $q(n)$ (called q-value) on $v(n)$. During search, RBFAOO improves and caches in a fixed size table $q(n)$ which is calculated by propagating back the q-values of $n$'s children. RBFAOO stops when $q(r) = v(r)$ at the root $r$ or it proves that there is no solution, namely $q(r) = v(r) = \infty$.

## 3   Our Parallel Algorithm

---
**Algorithm 1** SPRBFAOO
---
**for all** $i$ from 1 to nr CPU cores **do**
    $root.th \leftarrow \infty - \epsilon; root.thub \leftarrow \infty$
    launch tRBFS($root$) on a separate thread
wait for threads to finish their work
**return**  optimal cost (e.g., as root's $q$-value in the cache)

---

We now describe SPRBFAOO, a parallelization of RBFAOO in shared-memory environments. SPRBFAOO's threads start from the root and run in parallel, as shown in Algorithm 1. Threads share one cache table, allowing them to reuse the results of each other. An entry in the cache table, corresponding to a node $n$, is a tuple with 6 fields: a *q-value* $q(n)$, being a lower bound on the optimal cost of node $n$; $n.solved$, a flag indicating whether $n$ is solved optimally; a *virtual q-value* $vq(n)$, defined later in this section; a best known solution cost $bs(n)$ for node $n$; the number of threads currently working on $n$; and a lock. When accessing a cache entry, threads lock it temporarily for other threads. The method Ctxt($n$) identifies the context of $n$, which is further used to access the corresponding cache entry. Besides the cache, shared among threads, each thread will use two threshold values, $n.th$ and $n.thub$, for each node $n$. These are separated from one thread to another.

Algorithm 2 shows the procedure invoked on each thread. When a thread examines a node $n$, it first increments in the cache the number of threads working on node $n$ (line 1). Then it increases $vq(n)$ by an increment $\zeta$, and stores the new value in the cache (line 2). The virtual q-value $vq(n)$ is initially set to $q(n)$. As more threads work on solving $n$, $vq(n)$ grows due to the repeated increases by $\zeta$. In effect, $vq(n)$ reflects both the estimated cost of node $n$ (through its $q(n)$ component) and the number of threads working on $n$. By computing $vq(n)$ this way, our goal is to dynamically control the degree to which threads overlap when exploring the search space. When a given area of the search space is more promising than others, more than one thread are encouraged to work together within that area. On the other hand, when several areas are roughly equally promising, threads should diverge and work on different areas. Indeed, in Algorithm 2, the tests on lines 13 and 23 prevent a thread from working on a node $n$ if $n.th < vq(n)$. (Other conditions in these tests are discussed later.) A large $vq(n)$, which increases the likelihood that $n.th < vq(n)$, may reflect a less promising node (i.e., large q-value), or many threads working on $n$, or both. Thus, our strategy is an automated and dynamic way of tuning the number of threads working on solving a node $n$ as a function of how promising that node is. We call this the *thread coordination mechanism*.

Lines 4–7 address the case of nodes with no children, which are either terminal nodes or deadends. In both cases, method Evaluate sets the solved flag to true. The q-value $q$ is set to 0 for terminal

---

**Algorithm 2** Method tRBFS. Handling locks skipped for clarity.

---

**Require:** node $n$

1: IncrementNrThreadsInCache(Ctxt($n$))
2: IncreaseVQInCache(Ctxt($n$), $\zeta$))
3: **if** $n$ has no children **then**
4:     ($q$, $solved$) ← Evaluate($n$)
5:     SaveInCache(Ctxt($n$), $q$, $solved$, $q$, $q$)
6:     DecrementNrThreadsInCache(Ctxt($n$))
7:     **return**
8: GenerateChildren($n$)
9: **if** $n$ is an OR node **then**
10:    **loop**
11:        ($c_{best}$, $vq$, $vq_2$, $q$, $bs$) ← BestChild($n$)
12:        $n.thub$ ← min($n.thub$, $bs$)
13:        **if** $n.th < vq \vee q \geq n.thub \vee n.solved$ **then**
14:            **break**
15:        $c_{best}.th$ ← min($n.th$, $vq_2 + \delta$) − $w(n, c_{best})$
16:        $c_{best}.thub$ ← $n.thub$ − $w(n, c_{best})$
17:        tRBSF($c_{best}$)

18: [continued from previous column]
19: **if** $n$ is an AND node **then**
20:    **loop**
21:        ($q$, $vq$, $bs$) ← Sum($n$)
22:        $n.thub$ ← min($n.thub$, $bs$)
23:        **if** $n.th < vq \vee q \geq n.thub \vee n.solved$ **then**
24:            **break**
25:        ($c_{best}$, $q_{c_{best}}$, $vq_{c_{best}}$) ← UnsolvedChild($n$)
26:        $c_{best}.th$ ← $n.th$ − ($vq − vq_{c_{best}}$)
27:        $c_{best}.thub$ ← $n.thub$ − ($q − q_{c_{best}}$)
28:        tRBSF($c_{best}$)
29: **if** $n.solved$ $\vee$ NrThreadsCache(Ctxt($n$)) = 1
    **then**
30:    $vq$ ← $q$
31: DecrementNrThreadsInCache(Ctxt($n$))
32: SaveInCache(Ctxt($n$), $q$, $n.solved$, $vq$, $bs$)

---

nodes and to $\infty$ otherwise. Method SaveInCache takes as argument the context of the node, and four values to be stored in order in these fields of the corresponding cache entry: $q$, $solved$, $vq$ and $bs$.

Lines 10–17 and 20–28 show respectively the cases when the current node $n$ is an OR node or an AND node. Both these follow a similar high-level sequence of steps:

- Update $vq$, $q$, and $bs$ for $n$, from the children's values (lines 11, 21). Also update $n.thub$ (lines 12, 22), an upper bound for the best solution cost known for $n$ so far. Methods BestChild and Sum are shown in Algorithm 3. In these, child node information is either retrieved from the cache, if available, or initialized with an admissible heuristic function $h$.

- Perform the backtracking test (lines 13–14 and 23–24). The thread backtracks to $n$'s parent if at least one of the following conditions hold: $th(n) < vq(n)$, discussed earlier; $q(n) \geq n.thub$ i.e., a solution containing $n$ cannot possibly beat the best known solution (we call this the *suboptimality test*); or the node is solved. The solved flag is true iff the node cost has been proven to be optimal, or the node was proven not to have any solution.

- Otherwise, select a successor $c_{best}$ to continue with (lines 11, 25). At OR nodes $n$, $c_{best}$ is the child with the smallest $vq$ among all children not solved yet (see method BestChild). At AND nodes, any unsolved child can be chosen. Then, update the thresholds of $c_{best}$ (lines 15–16 and 26–27), and recursively process $c_{best}$ (lines 17, 28). The threshold $n.th$ is updated in a similar way to RBFAOO, including the overestimation parameter $\delta$ (see [2]). However, there are two key differences. First, we use $vq$ instead of $q$, to obtain the thread coordination mechanism presented earlier. Secondly, we use two thresholds, $th$ and $thub$, instead of just $th$, with $thub$ being used to implement the suboptimality test $q(n) \geq n.thub$.

When a thread backtracks to $n$'s parent, if either $n$'s solved flag is set, or no other thread currently examines $n$, the thread sets $vq(n)$ to $q(n)$ (lines 29–30 in Algorithm 2). In this way, SPRBFAOO reduces the frequency of the scenarios where $n$ is considered to be less promising. Finally, the thread decrements in the cache the number of threads working on $n$ (line 31), and saves in the cache the recalculated $vq(n)$, $q(n)$, $bs(n)$, and the solved flag (line 32).

**Theorem 3.1.** *With an admissible heuristic in use, SPRBFAOO returns optimal solutions.*

*Proof sketch.* SPRBFAOO's $bs(r)$ at the root $r$ is computed from a solution tree, therefore, $bs(r) \geq v(r)$. Additionally, SPRBFAOO determines solution optimality by using not $vq(n)$ but $q(n)$ saved in the cache table. By an induction-based discussion similar to Theorem 3.1 in [2], $q(n) \leq v(n)$ holds for any $q(n)$ saved in the cache table with admissible $h$, which indicates $q(r) \leq v(r)$. When SPRBFAOO returns a solution, $bs(r) = q(r)$, therefore, $bs(r) = q(r) = v(r)$.  □

We conjecture that SPRBFAOO is also complete, and leave a more in-depth analysis as future work.

**Algorithm 3** Methods BestChild (left) and Sum (right)

**Require:** node $n$                                    **Require:** node $n$
  1: $n.solved \leftarrow \bot$ ($\bot$ stands for **false**);       1: $n.solved \leftarrow \top$ ($\top$ stands for **true**)
  2: initialize $vq, vq_2, q, bs$ to $\infty$               2: initialize $vq, q, bs$ to $0$
  3: **for all** $c_i$ child of $n$ **do**                   3: **for all** $c_i$ child of $n$ **do**
  4:   **if** Ctxt$(c_i)$ in cache **then**                  4:   **if** Ctxt$(c_i)$ in cache **then**
  5:     $(q_{c_i}, s_{c_i}, vq_{c_i}, bs_{c_i}) \leftarrow$ FromCache(Ctxt$(c_i)$)   5:     $(q_{c_i}, s_{c_i}, vq_{c_i}, bs_{c_i}) \leftarrow$ FromCache(Ctxt$(c_i)$)
  6:   **else**                                             6:   **else**
  7:     $(q_{c_i}, s_{c_i}, vq_{c_i}, bs_{c_i}) \leftarrow (h(c_i), \bot, h(c_i), \infty)$   7:     $(q_{c_i}, s_{c_i}, vq_{c_i}, bs_{c_i}) \leftarrow (h(c_i), \bot, h(c_i), \infty)$
  8:   $q_{c_i} \leftarrow w(n, c_i) + q_{c_i}$              8:   $q \leftarrow q + q_{c_i}$
  9:   $vq_{c_i} \leftarrow w(n, c_i) + vq_{c_i}$            9:   $vq \leftarrow vq + vq_{c_i}$
 10:   $bs = \min(bs, w(n, c_i) + bs_{c_i})$               10:   $bs \leftarrow bs + bs_{c_i}$
 11:   **if** $(q_{c_i} < q) \vee (q_{c_i} = q \wedge \neg n.solved)$ **then**   11:   $n.solved \leftarrow n.solved \wedge s_{c_i}$
 12:     $n.solved \leftarrow s_{c_i}; q \leftarrow q_{c_i}$   12: **return** $(q, vq, bs)$
 13:   **if** $vq_{c_i} < vq \wedge \neg s_{c_i}$ **then**
 14:     $vq_2 \leftarrow vq; vq \leftarrow vq_{c_i}; c_{best} \leftarrow c_i$
 15:   **else if** $vq_{c_i} < vq_2 \wedge \neg s_{c_i}$ **then**
 16:     $vq_2 \leftarrow vq_{c_i}$
 17: **return** $(c_{best}, vq, vq_2, q, bs)$

## 4 Experiments

We evaluate empirically our parallel SPRBFAOO and compare it against sequential RBFAOO and AOBB. We also considered *parallel shared-memory* AOBB, denote by SPAOBB, which uses a master thread to explore centrally the AND/OR search graph up to a certain depth and solves the remaining conditioned sub-problems in parallel using a set of worker threads. The cache table is shared among the workers so that some workers may reuse partial search results recorded by others. In our implementation, the search space explored by the master corresponds to the first $m$ variables in the pseudo tree. The performance of SPAOBB was very poor across all benchmarks due to noticeably large search overhead as well as poor load balancing, and therefore its results are omitted hereafter.

All competing algorithms (SPRBFAOO, RBFAOO and AOBB) use the pre-compiled *mini-bucket heuristic* [1] for guiding the search. The heuristic is controlled by a parameter called $i$-bound which allows a trade-off between accuracy and time/space requirements – higher values of $i$ yield a more accurate heuristic but take more time and space to compute. The search algorithms were also restricted to a static variable ordering obtained as a depth-first traversal of a min-fill pseudo tree [1].

Our benchmark problems[1] include three sets of instances from genetic linkage analysis (denoted `pedigree`) [14], grid networks and protein side-chain interaction networks (denoted `protein`) [15]. In total, we evaluated 21 pedigrees, 32 grids and 240 protein networks. The algorithms were implemented in C++ (64-bit) and the experiments were run on a 2.6GHz 12-core processor with 80GB of RAM. Following [2], RBFAOO ran with a 10-20GB cache table (134,217,728 entries) and overestimation parameter $\delta = 1$. However, SPRBFAOO allocated only 95,869,805 entries with the same amount of memory, due to extra information such as virtual q-values. We set $\zeta = 0.01$ throughout the experiments (except those where we vary $\zeta$). The time limit was set to 2 hours. We also record typical ranges of problem specific parameters shown in Table 1 such as the number of variables ($n$), maximum domain size ($k$), induced width ($w^*$), and depth of the pseudo tree ($h$).

Table 1: Ranges (min-max) of the benchmark problems parameters.

| benchmark | $n$ | $k$ | $w^*$ | $h$ |
|---|---|---|---|---|
| grid | $144 - 676$ | $2$ | $15 - 36$ | $48 - 136$ |
| pedigree | $334 - 1289$ | $3 - 7$ | $15 - 33$ | $51 - 140$ |
| protein | $26 - 177$ | $81$ | $6 - 16$ | $15 - 43$ |

Table 2: Number of unsolved problem instances (1 vs 12 cores).

| method | grid | | pedigree | | protein | |
|---|---|---|---|---|---|---|
| | $i = 6$ | $i = 14$ | $i = 6$ | $i = 14$ | $i = 2$ | i=4 |
| RBFAOO | 9 | 5 | 8 | 6 | 41 | 16 |
| SPRBFAOO | 7 | 5 | 7 | 3 | 32 | 9 |

The primary performance measures reported are the run time and node expansions during search. When the run time of a solver is discussed, the total CPU time reported in seconds is one metric to show overall performance. The total CPU time consists of the heuristic compilation time and search

Table 3: Total CPU time (sec) and nodes on `grid` and `pedigree` instances. Time limit 2 hours.

| instance $(n,k,w^*,h)$ | algorithm | $i=6$ time | $i=6$ nodes | $i=8$ time | $i=8$ nodes | $i=10$ time | $i=10$ nodes | $i=12$ time | $i=12$ nodes | $i=14$ time | $i=14$ nodes |
|---|---|---|---|---|---|---|---|---|---|---|---|
| **75-22-5** (484,2,30,107) | (mbe) | (0.06) | | (0.07) | | (0.1) | | (0.2) | | (0.7) | |
| | AOBB | | | | | 5221 | 761867041 | 2100 | 314622599 | 884 | 144092486 |
| | RBFAOO | 629 | 133143216 | 2018 | 331885596 | 2036 | 334441548 | 638 | 113597702 | 85 | 18728991 |
| | SPRBFAOO | **116** | 153612683 | **483** | 410230906 | **466** | 385071090 | **152** | 129817500 | **17** | 25076772 |
| **75-24-5** (576,2,32,116) | (mbe) | (0.08) | | (0.1) | | (0.1) | | (0.3) | | (0.8) | |
| | AOBB | | | | | | | | | | |
| | RBFAOO | | | | | 4182 | 665237411 | 2792 | 465384385 | 229 | 47015068 |
| | SPRBFAOO | **2794** | 2273916962 | **2959** | 2309390159 | **1012** | 804068930 | **579** | 511894256 | **43** | 59504303 |
| **90-30-5** (900,2,42,151) | (mbe) | (0.2) | | (0.2) | | (0.3) | | (0.5) | | (1.4) | |
| | AOBB | | | | | | | | | | |
| | RBFAOO | | | | | | | | | 3783 | 565053698 |
| | SPRBFAOO | | | | | | | | | **869** | 665947009 |
| **pedigree7** (1068,4,28,140) | (mbe) | (0.1) | | (0.2) | | (0.3) | | (0.6) | | (2.1) | |
| | AOBB | | | | | | | | | | |
| | RBFAOO | | | | | 1873 | 226436502 | 1642 | 201063828 | 1239 | 135387634 |
| | SPRBFAOO | | | **4560** | 3062954989 | **353** | 249562472 | **314** | 222896697 | **267** | 151794050 |
| **pedigree9** (1119,7,25,123) | (mbe) | (0.1) | | (0.2) | | (0.2) | | (0.5) | | (1.6) | |
| | AOBB | | | | | | | | | | |
| | RBFAOO | | | | | | | | | | |
| | SPRBFAOO | | | | | | | | | 3021 | 2807834881 |
| **pedigree19** (793,5,21,107) | (mbe) | (0.1) | | (0.2) | | (0.4) | | (1.3) | | (10) | |
| | AOBB | | | | | | | | | | |
| | RBFAOO | | | | | | | | | | |
| | SPRBFAOO | | | | | | | **3792** | 2721253097 | **2083** | 1914585138 |

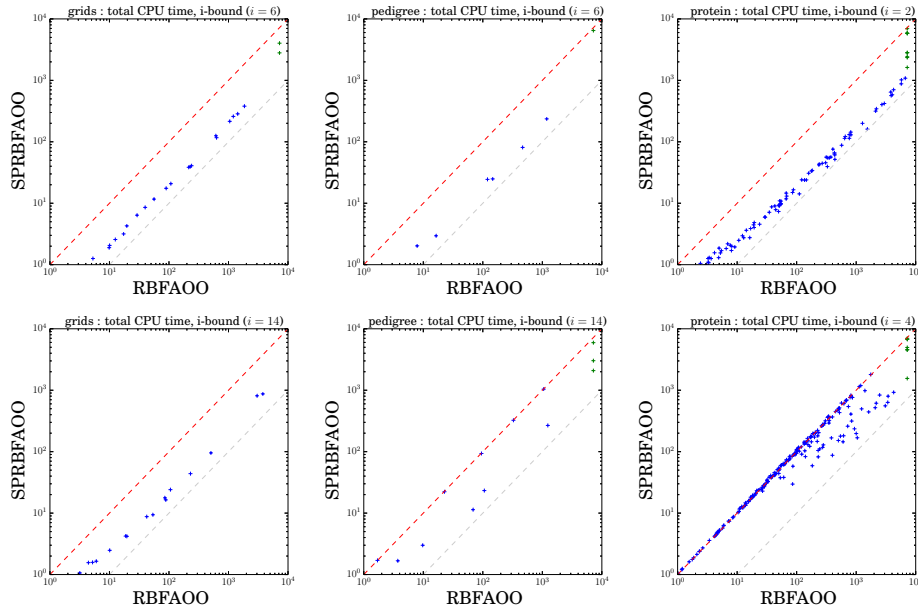

Figure 2: Total CPU time (sec) for RBFAOO vs. SPRBFAOO with smaller (top) and larger (bottom) $i$-bounds. Time limit 2 hours. $i \in \{6, 14\}$ for `grid` and `pedigree`, $i \in \{2, 4\}$ for `protein`.

time. SPRBFAOO does not reduce the heuristic compilation time calculated sequentially. Note that parallelizing the heuristic compilation is an important extension as future work.

**Parallel versus sequential search** Table 3 shows detailed results (as total CPU time in seconds and nodes expanded) for solving `grid` and `pedigree` instances using parallel and sequential search. The columns are indexed by the $i$-bound. For each problem instance, we also record the mini-bucket heuristic pre-compilation time, denoted by (mbe), corresponding to each $i$-bound. SPRBFAOO ran with 12 threads. We can see that SPRBFAOO improves considerably over RBFAOO across all reported $i$-bounds. The benefit of parallel search is more clearly observed at smaller $i$-bounds that correspond to relatively weak heuristics. In this case, the heuristic is less likely to guide the search towards more promising regions of the search space and therefore diversifying the search via multiple parallel threads is key to achieving significant speed-ups. For example, on grid 75-22-5, SPRBFAOO(6) is almost 6 times faster than RBFAOO(6). Similarly, SPRBFAOO(8) solves the pedigree7 instance while RBFAOO(8) runs out of time. This is important since on very hard problem instances it may only be possible to compute rather weak heuristics given limited resources. Notice

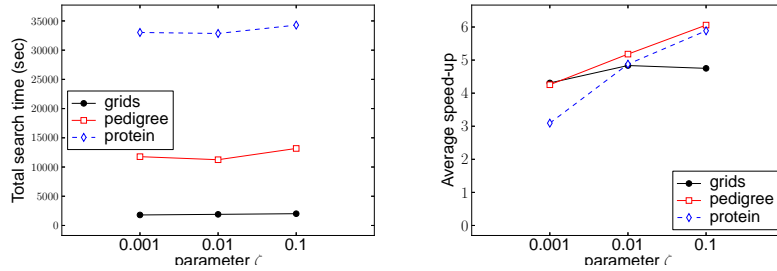

Figure 3: Total search time (sec) and average speed-up as a function of parameter $\zeta$. Time limit 2 hours. $i = 14$ for `grid` and `pedigree`, $i = 4$ for `protein`.

also that the pre-processing time (mbe) increases with the $i$-bound. Table 2 shows the number of *unsolved* problems in each domain. Note that SPRBFAOO solved all instances solved by RBFAOO.

Figure 2 plots the total CPU time obtained by RBFAOO and SPRBFAOO using smaller (resp. larger) $i$-bounds corresponding to relatively weak (resp. strong) heuristics. We selected $i \in \{6, 14\}$ for `grid` and `pedigree`, and $i \in \{2, 4\}$ for `protein`. Specifically, $i = 6$ (grids, pedigrees) and $i = 2$ (proteins) were the smallest $i$-bounds for which SPRBFAOO could solve at least two thirds of instances within the 2 hour time limit, while $i = 14$ (grids, pedigrees) and $i = 4$ (proteins) were the largest possible $i$-bounds for which we could compile the heuristics without running out of memory on *all* instances. The data points shown in green correspond to problem instances that were solved only by SPRBFAOO. As before, we notice the benefit of parallel search when using relatively weak heuristics. The largest speed-up of 9.59 is obtained on the pdb1kl `protein` instance with $i = 2$. As the $i$-bound increases and the heuristics become more accurate, the difference between RBFAOO($i$) and SPRBFAOO($i$) decreases because both algorithms are guided more effectively towards the subspace containing the optimal solution. In addition, the overhead associated with larger $i$-bounds, which is calculated sequentially, offsets considerably the speed-up obtained by SPRBFAOO($i$) over RBFAOO($i$) (see for example the plot for `protein` instances with $i = 4$).

We also observed that SPRBFAOO's speed-up over RBFAOO increases sublinearly as more threads are used (we experimented with 3, 6, and 12 threads, respectively). In addition to search overhead, synchronization overhead is another cause for achieving only sublinear speed-ups. The synchronization overhead can be estimated by checking the node expansion rate per thread. For example, in case of SPRBFAOO with 12 threads, the node expansion rate per thread slows down to 47 %, 50 %, and 61 % of RBFAOO in `grid` ($i = 6$), `pedigree` ($i = 6$), and `protein` ($i = 2$), respectively. This implies that the overhead related to locks is large. Since these numbers with 6 threads are 73 %, 79 %, and 96 %, respectively, the slowdown becomes severer with more threads. We hypothesize that due to the property of the virtual q-value, SPRBFAOO's threads tend to follow the same path from the root until search directions are diversified, and frequently access the cache table entries of the these internal nodes located on that path, where lock contentions occur non-negligibly.

Finally, SPRBFAOO's load balance is quite stable in all domains, especially when all threads are invoked and perform search after a while. For example, its load balance ranges between 1.005-1.064, 1.013-1.049, and 1.004-1.117 for `grid` ($i = 6$), `pedigree` ($i = 6$), and `protein` ($i = 2$), especially on those instances where SPRBFAOO expands at least 1 million nodes with 12 threads.

**Impact of parameter** $\zeta$  In Figure 3 we analyze the performance of SPRBFAOO with 12 threads as a function of the parameter $\zeta$ which controls the way different threads are encouraged or discouraged to start exploring a specific subproblem (see also Section 3). For this purpose and to better understand SPRBFAOO's scaling behavior, we ignore the heuristic compilation time. Therefore, we show the total search time (in seconds) over the instances that all parallel versions solve, and the search-time-based average speed-ups based on the instances where RBFAOO needs at least 1 second to solve. We obtained these numbers for $\zeta \in \{0.001, 0.01, 0.1\}$. We see that all $\zeta$ values lead to improved speed-ups. This is important because, unlike the approach of [8] which involves a sophisticated scheme, it is considerably simpler yet extremely efficient and only requires tuning a single parameter ($\zeta$). Of the three $\zeta$ values, while SPRBFAOO with $\zeta = 0.1$ spends the largest total search time, it yields the best speed-up. This indicates a trade-off about selecting $\zeta$. Since the instances used to calculate speed-up values are solved by RBFAOO, they contain relatively easy instances.

Table 4: Total CPU time (sec) and node expansions for hard pedigree instances. SPRBFAOO ran with 12 threads, $i = 20$ (type4b) and $i = 16$ (largeFam). Time limit 100 hours.

| instance | $(n, k, w^*, h)$ | (mbe) time | RBFAOO time | nodes | SPRBFAOO time | nodes |
|---|---|---|---|---|---|---|
| type4b-100-19 | (7308,5,29,354) | 400 | 132711 | 22243047591 | **42846** | 50509174040 |
| type4b-120-17 | (7766,5,24,319) | 191 | 210 | 4297063 | **195** | 6046663 |
| type4b-130-21 | (8883,5,29,416) | 281 | 290760 | 51481315386 | **149321** | 177393525747 |
| type4b-140-19 | (9274,5,30,366) | 488 | 248376 | 39920187143 | **74643** | 85152364623 |
| largeFam3-10-52 | (1905,3,36,80) | 13 | 154994 | 19363865449 | **50700** | 44073583335 |

On the other hand, several difficult instances solved by SPRBFAOO with 12 threads are included in calculating the total search time. In case of $\zeta = 0.1$, because of increased search overhead, SPRBFAOO needs more search time to solve these difficult instances. There is also one `protein` instance unsolved with $\zeta = 0.1$ but solved with $\zeta = 0.01$ and $0.001$. This phenomenon can be explained as follows. With large $\zeta$, SPRBFAOO searches in more diversified directions which could reduce lock contentions, resulting in improved speed-up values. However, due to larger diversification, when SPRBFAOO with $\zeta = 0.1$ solves difficult instances, it might focus on less promising portions of the search space, resulting in increased total search time.

**Summary of the experiments** In terms of search-time-based speed-ups, our parallel shared-memory method SPRBFAOO improved considerably over its sequential counterpart RBFAOO, by up to 7 times using 12 threads. At relatively larger $i$-bounds, their corresponding computational overhead typically outweighed the gains obtained by parallel search. Still, parallel search had an advantage of solving additional instances unsolved by serial search. Finally, in Table 4 we report the results obtained on 5 very hard pedigree instances from [2] (mbe records the heuristic compilation time). We see again that SPRBFAOO improved over RBFAOO on all instances, while achieving a total-time-based speed-up of 3 on two of them (i.e., type4b-100-19 and largeFam3-10-52).

## 5 Related Work

The distributed AOBB algorithm `daoopt` [8] which builds on the notion of parallel tree search [16], explores centrally the search tree up to a certain depth and solves the remaining conditioned sub-problems in parallel using a large grid of distributed processing units without a shared cache.

In parallel evidence propagation, the notion of *pointer jumping* has been used for exact probabilistic inference. For example, Pennock [17] performs a theoretical analysis. Xia and Prasanna [18] split a junction tree into chains where evidence propagation is performed in parallel using a distributed-memory environment, and the results are merged later on.

Proof-number search (PNS) in AND/OR spaces [19] and its parallel variants [20] have been shown to be effective in two-player games. As PNS is suboptimal, it cannot be applied as is to exact MAP inference. Kaneko [21] presents shared-memory parallel depth-first proof-number search with *virtual proof and disproof numbers (vpdn)*. These combine proof and disproof numbers [19] and the number of threads examining a node. Thus, our $vq(n)$ is closely related to vpdn. However, vpdn has an over-counting problem, which we avoid due to the way we dynamically update $vq(n)$. Saito et al. [22] uses threads that probabilistically avoid the best-first strategy. Hoki et al. [23] adds small random values the proof and disproof numbers of each thread without sharing any cache table.

## 6 Conclusion

We presented SPRBFAOO, a new shared-memory parallel recursive best-first AND/OR search scheme in graphical models. Using the virtual q-values shared in a single cache table, SPRBFAOO enables threads to work on promising regions of the search space with effective reuse of the search effort performed by others. A homogeneous search mechanism across the threads achieves an effective load balancing without resorting to sophisticated schemes used in related work [8]. We prove the correctness of the algorithm. In experiments, SPRBFAOO improves considerably over current state-of-the-art sequential AND/OR search approaches, in many cases leading to considerable speed-ups (up to 7-fold using 12 threads) especially on hard problem instances. Ongoing and future research directions include proving the completeness conjecture, extending SPRBFAOO to distributed memory environments, and parallelizing the mini-bucket heuristic for shared and distributed memory.

## Footnotes

[1]*http://graphmod.ics.uci.edu*

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
