[Reviews · NeurIPS 2015]

Submitted by Assigned_Reviewer_1

* Summary

This paper contributes a concurrent recursive best-first search algorithm to the class of AND/OR search approaches to discrete optimization, a proof of its correctness and a convincing experimental assessment.

* Quality

This paper of of high quality. The algorithm design is convincing. The parallelization strategy is a non-trivial and novel contribution to AND/OR search approaches to discrete optimization. The proof of correctness appears to be sound. The experimental results demonstrate convincingly that the parallelization strategy is beneficial within certain bounds.

The paper could be strengthened further by comparing the proposed AND/OR search algorithm also with polyhedral optimization methods, in particular, integer programming.

* Clarity

The paper is well-written. I particularly like to accurate informal description of AND/OR search which makes the paper sufficiently self-contained.

* Originality

The parallelization strategy is a non-trivial and novel contribution to AND/OR search approaches to discrete optimization.

* Significance

Although this paper would be suitable also for an algorithms/high-performance computing conference, its original contribution is relevant for MAP inference in graphical models and is thus of interest to a large part of the NIPS community.

Summary: This well-written, high-quality paper contributes a concurrent recursive best-first search algorithm to the class of AND/OR search approaches to discrete optimization, a proof of its correctness and a convincing experimental assessment. Although this paper would be suitable also for an algorithms/high-performance computing conference, its original contribution is relevant for MAP inference in graphical models and is thus of interest to a large part of the NIPS community.

Submitted by Assigned_Reviewer_2

The paper proposes a parallelization for an important combinatorial problem in probabilistic graphical models, namely to find the full assignment of maximum probability (called MAP in this work). The proposal is conceptually simple, but the improvement over the serial version might be useful in practice. The paper is very well written and presented. Parallel methods for this problem are still lacking, so this is a valuable contribution. My only concern regards the comparison against other state-of-the-art methods for MAP. The paper does not seem to cite some recent attempts to tackle this problem, and I believe that some of them could be included in the evaluation (there are multiple packages available online for MAP, see e.g. UAI 2014 competition, and also MPLP, AD3, libdai). My reasoning is that the parallelization improvement that is brought to RBFAOO is only useful in practice if RBFAOO performs well over the test cases that were analysed in this work when compared to other serial methods. As the contribution here is mostly practical, such comparisons are very important.
Summary: Nice contribution from a practical point of view about the parallelization of a particular algorithm (RBFAOO) for MAP in probabilistic graphical models. Lacks comparisons to indicate that parallelizing RBFAOO is the way to go, instead of going for other state-of-the-art methods.

Submitted by Assigned_Reviewer_3

The paper is straightforward in a good way. RBFAOO is briefly described and then an appropriately detailed description of SPRBFAOO is given. It is clear that some thought (and no doubt experimentation) has gone into the presented design. As it should be, there is quite a long section on empirical results which help the reader understand the important issues.

It would be interesting to see whether a "programming by optimisation (PBO)" approach would yield good results - perhaps eg fitting \zeta to features of a particular problem instance.

In [2] RBFAOO is only compared against AOBB and AOBF, so although it is superior to both those algorithms it is not clear to me how it compares to alternative graphical model MAP algorithms such as those which participated in the UAI'14 inference challenge (which had a MAP competition). Note that the proteus solver, a portfolio solver, did well (beating daopt in the 20 and 60 min competitions). It included CPLEX which automatically parallelises if there are the processors available, so that's another example of parallel search for MAP. I acknowledge that the parallelisation strategy here is more sophisticated and problem-specfic than sending the problem to a general-purpose solver that happens to be able to run in parallel.

I suspect this paper would be of interest to a fairly narrow pool of NIPS people since it has a narrow focus. It is no surprise that, if done competently, using more computational power can speed up (a particular approach to )MAP solving in MRFs.

typos, etc:

reduces the frequency of occurring the scenarios -> reduces the frequency of scenarios
Summary: The paper introduces SPRBFAOO a shared-memory parallelisation of the RBFAOO algorithm. It is shown that there are good speed-ups to be had if the threads are available.

Submitted by Assigned_Reviewer_4

The authors combine recursive best-first search with a scheme for organizing exploration among multiple cores in a shared-memory architecture.

This paper is very well-written and clear, with good algorithmic descriptions of the method.

It also appears to work reasonably well, with speedup factors of ~7 on a 12-core architecture.

My only concern is that its main contribution is implementation; the algorithmic contribution seems relatively small (mainly, the use of vq(n) and gamma to organize the exploration across threads), and it is hard for me to judge the significance of that element.

However, both MAP solvers and multi-core parallelism are topics that are important to many at NIPS.

One point the authors could perhaps explore further is, when the node expansion rate is low and locking becomes a significant overhead, whether there are simple modifications to the search procedure which could ensure less locking, for example using a small depth-first expansion at each best-first node to reduce the locking rate.
Summary: A well-written, practical contribution to parallel shared-memory MAP search; the main weakness is that it is mostly a simple combination of existing ideas, with one of the major contributions being the implementation and testing of that combination.

Author Feedback
Author rebuttal: We thank the reviewers for the detailed comments and feedback. In the following we will address some of the issues that have been raised by the reviewers.

1) Comparison with state-of-the-art (reviewers 2,3,4)

We would like to emphasize that the focus of this paper is on *exact* MAP inference. Current state-of-the-art is represented by search based algorithms guided by either mini-bucket heuristics (daoopt, AOBB) or soft local consistency (toulbar). These algorithms were compared against each other in the past 2-4 UAI competitions and were shown to be both winning and overall both quite competitive. Since we compare with one of these schemes we believe we compare against state-of-the-art for exact MAP inference. For example, to the best of our knowledge, the type4/largeFam instances shown in Table 4 are solved only by RBFAOO/SPRBFAOO with the mini-bucket heuristics. The toulbar solver cannot optimally solve many of the pedigree instances including pedigree 7, 9 and 19 shown in Table 3. Moreover, using the same mini-bucket heuristic as RBFAOO/AOBB as in the literature, allows us to evaluate our parallel search scheme as fairly as possible against sequential search. Finally, since our approach uses a general purpose heuristic (ie, mini-buckets) it can be extended relatively easily to harder tasks such as Marginal MAP or maximum expected utility queries for which toulbar as well as CPLEX algorithms are not feasible.

MPLP as well as most of the algorithms available from libDAI are *approximate* algorithms and cannot guarantee optimality in general. Comparing against these classes of approximate algorithms would be somewhat misleading since our focus is on exact algorithms. However, parallelizing these kinds of message passing approximate algorithms is definitely a good direction for future work. In fact, our current work is centered on parallelizing the mini-bucket algorithm which is the basis of our heuristic and can also be viewed as a message passing algorithm.

We believe that portfolio solvers such as proteus are actually orthogonal to our approach in the sense that SPRBFAOO can always be part of such a solver. Moreover, one of the core solvers of proteus is toulbar which was shown to be competitive with AOBB during past UAI competitions. Comparison with CPLEX on a 0-1 ILP encoding of the graphical model is definitely called for. However we think this would be outside the scope of the current paper and more appropriate for future work because RBFAOO and SPRBFAOO can also be used to solve 0-1 ILPs in the same way AOBB and AOBF were extended in the past to solve 0-1 ILPs (see also R. Marinescu and R. Dechter. Evaluating the impact of AND/OR search on 0-1 integer linear programming. In Constraints 15(1):29-63, 2010.). Furthermore, since CPLEX is a commercial product we don't have enough details on the parallelization scheme employed by CPLEX and therefore it would be hard to draw meaningful conclusions when comparing with our approach.

2) Locking overhead (reviewer 5)

Locking tends to incur a significant overhead when the expansion rate is high (ie, high number of node expansions per second). The locking overhead is still an open problem in general. In our experiments this overhead was almost negligible. We implemented a fast spin lock with an x86 assembly language operation (see the first paragraph of Section 5 in the supplementary material). However, there are methods that aim at minimizing the locking overhead, such as using lock-free data structures. We plan to investigate these approaches especially when we move beyond 12 cores.

3) Naive parallelization scheme (reviewer 6)

We actually implemented and experimented with the naive parallelization scheme suggested using AOBB to solve the conditioned energy minimization subproblems (we called it SPAOBB). It performed very poorly and therefore was excluded from the main paper due to space reasons (see also 1st paragraph of Section 4). We will include these results in a long version of the paper.

4) Pseudo-code (reviewer 4)

We adopted a C++ like pseudo-code in order to be easier for a reader to actually implement the algorithm and duplicate the results. We will consider simplifying the pseudo-code.